Innovative infrastructure to access Brazilian fungal diversity using deep learning

http://orcid.org/0000-0002-1737-8806 Chaves Thiago 1
http://orcid.org/0000-0002-4649-6270 Santos Xavier Joicymara 2
Gonçalves dos Santos Alfeu 1
http://orcid.org/0000-0002-0545-5966 Martins-Cunha Kelmer 3
Karstedt Fernanda 3
Kossmann Thiago 3
Sourell Susanne 3
http://orcid.org/0000-0002-6060-7731 Leopoldo Eloisa 3
Fortuna Ferreira Miriam Nathalie 1
Farias Roger 1
http://orcid.org/0000-0001-7139-5116 Titton Mahatmã 3
http://orcid.org/0000-0002-8142-6665 Alves-Silva Genivaldo 3
Bittencourt Felipe 3
Bortolini Dener 4
http://orcid.org/0000-0002-1237-2050 Gumboski Emerson L. 5
http://orcid.org/0000-0003-4532-1417 von Wangenheim Aldo 1
http://orcid.org/0000-0002-7692-6243 Góes-Neto Aristóteles 4 arigoesneto@gmail.com
Drechsler-Santos Elisandro Ricardo 3
1 Brazilian National Institute for Digital Convergence—INCoD, Universidade Federal de Santa Catarina , Florianópolis, Santa Catarina , Brazil
2 Institute of Agricultural Science, Universidade Federal dos Vales do Jequitinhonha e Mucuri , Unaí, Minas Gerais , Brazil
3 MIND.Funga/MICOLAB, Department of Botany, Universidade Federal de Santa Catarina , Florianópolis, Santa Catarina , Brazil
4 Department of Microbiology, Institute of Biological Sciences, Universidade Federal de Minas Gerais (UFMG) , Belo Horizonte, Minas Gerais , Brazil
5 Department of Biological Sciences, Regional University of Joinville (UNIVILLE) , Joinville, Santa Catarina , Brazil
Gillespie Joseph
Electronic publication date: 2024 Jul 9
Publication date: 2024
Volume: 12
Electronic Location ID: e17686
Received 2024 Jan 19; Accepted 2024 Jun 13
Copyright: © 2024 Chaves et al.
Copyright year: 2024
Copyright holder: Chaves et al.
License: This is an open access article distributed under the terms of the Creative Commons Attribution License, which permits unrestricted use, distribution, reproduction and adaptation in any medium and for any purpose provided that it is properly attributed. For attribution, the original author(s), title, publication source (PeerJ) and either DOI or URL of the article must be cited.
License URL: https://creativecommons.org/licenses/by/4.0/

Keywords: Deep learning, Computer vision, CNN, Image classification, Fungi

Funding: CNPq 310150/2022-1, 153025/2022-0 Conselho Nacional de Desenvolvimento Científico e Tecnológico Coordenação de Aperfeiçoamento Pessoal de Nível Superior (CAPES) Mohamed bin Zayed Species Conservation Fund (MBZ) N° 222530589, 232531493 Elisandro R. Drechsler-Santos (CNPq, grant no. 310150/2022-1) and Genivaldo Alves-Silva (CNPq, grant no. 153025/2022-0) were supported by Conselho Nacional de Desenvolvimento Científico e Tecnológico. Felipe Bittencourt and Thiago Kossmann were supported with scholarships by the Coordenação de Aperfeiçoamento Pessoal de Nível Superior (CAPES), and Kelmer Martins-Cunha (N° 222530589 and 232531493) by The Mohamed bin Zayed Species Conservation Fund (MBZ). The funders had no role in study design, data collection and analysis, decision to publish, or preparation of the manuscript.

==============================
In the present investigation, we employ a novel and meticulously structured database assembled by experts, encompassing macrofungi field-collected in Brazil, featuring upwards of 13,894 photographs representing 505 distinct species. The purpose of utilizing this database is twofold: firstly, to furnish training and validation for convolutional neural networks (CNNs) with the capacity for autonomous identification of macrofungal species; secondly, to develop a sophisticated mobile application replete with an advanced user interface. This interface is specifically crafted to acquire images, and, utilizing the image recognition capabilities afforded by the trained CNN, proffer potential identifications for the macrofungal species depicted therein. Such technological advancements democratize access to the Brazilian Funga, thereby enhancing public engagement and knowledge dissemination, and also facilitating contributions from the populace to the expanding body of knowledge concerning the conservation of macrofungal species of Brazil.

Introduction

Fungi are ubiquitous organisms that play vital roles in virtually all terrestrial ecosystems. Their diversity is represented by the term Funga, equivalently to Fauna and Flora, which represent the diversity of animals and plants respectively (Kuhar et al., 2018). The kingdom Fungi is one of the most diverse taxonomic groups, with mainly accepted estimates currently at 2.5 million species (Niskanen et al., 2023). Nevertheless, only a fraction of the Funga is known, since less than 10% of these species are described (Antonelli et al., 2020).

Fungi act as key nutrient recyclers and form essential mutualistic relationships with several groups (Watkinson, Boddy & Money, 2016). Therefore, their ecological importance contrasts with this vast knowledge gap. In addition, the slow process of species discovery and knowledge construction worsens the Funga neglected scenario. It can take decades to build well-founded knowledge regarding plant taxa (Haridas et al., 2020), a field that arguably receives more attention than Mycology.

Hence, there is an urge to find ways to add up efforts and fasten the process of describing the fungal species. The lack of knowledge about Funga reflects the negative relationship between the general Brazilian public and fungi, which can be considered mainly mycophobic (Góes-Neto & Bandeira, 2003). Part of this repulsion to fungi could be attributed to their role in decaying food and causing diseases (Irga, Barker & Torpy, 2018), coupled with the lack of awareness about their ecological importance.

Furthermore, Funga has its significance downplayed even in higher education, with Mycology frequently not being treated as a course, but as a section in related grades (Grube et al., 2017). As the Funga is neglected and constitutes a key element in ecosystem health, more efforts must be made to fill the knowledge gap in fungal diversity while making the general public aware of its utter importance, as general awareness and affinity are positively correlated with greater incentive for biodiversity protection (Awasthy, Popovic & Linklater, 2012; Beery & Jørgensen, 2016).

The enrolment of the general public in efforts to access the Funga (e.g., citizen science programs) has been proven to be effective in species detection and discovery (Heilmann-Clausen et al., 2019; Crous et al., 2021; Hou et al., 2020), reaching much higher sampling efforts than specialists can do (Heilmann-Clausen et al., 2016). Specifically, the use of innovative techniques based on widely available strategies can speed up this process, such as online education programs, interactive exhibitions, and applications that benefit from cutting-edge technologies and algorithms (Grassini, 2023), such as computer vision (CV) and deep learning (DL).

The general principle behind the computer vision paradigm is that an image interpretation task starts from meaningless pixels and moves stepwise toward a meaningful representation of that image content (Duda, Hart & Stork, 2001). Computer vision approaches can be roughly divided into classic computer vision (CCV) and deep learning-based computer vision (DL-CV).

CCV are performed as a pipeline of steps (Danuser, 2011), transforming the original images and adding different levels of abstraction. The first transform in this pipeline focuses on filtering and separating regions of interest (ROIs or foreground pixels) from background pixels (I->I), mainly using digital image processing methods such as noise filters, border detectors, and segmentation algorithms (Duda, Hart & Stork, 2001). Subsequently, when the abstraction level grows, these transforms are performed from images into models (I→M) that represent descriptions of specific elements or the content of these images, such as segment color, shape parameters, texture descriptors, and mathematical and statistics models.

Posteriorly, these models are transformed into more abstract models (M→M) that describe the meaning or classification of those objects, generally using a group of rules modeling some desirable traits to perform the task (Duda, Hart & Stork, 2001). These rules are strongly parameter-dependent and make these CCV pipelines extremely problem- and image-type-specific constrained (Uchida, 2013), meaning that a processing pipeline that works well for a given type of image content, e.g., classification of tree crowns, will not work for agricultural weed detection. Often, the sensitivity of the parameters is so high that even lightning and contrast differences caused by an outdoor ambient light in the same set of images can be hard to tackle, requiring deep mathematical knowledge of the individual methods or real-world solutions (e.g., acquire images at the same time of the day or night, or yet, building some physical apparatus) to solve the problems. These characteristics make CCV solutions less robust and highly image-quality-dependent, which has, for a long time, limited the popularization of CV software for biological sciences as a whole, and more specifically, for Mycology.

The advent of DL techniques has broken the CCV paradigm. Initially proposed by LeCun et al. (1998) and popularized by Krizhevsky, Sutskever & Hinton (2012), DL convolutional neural networks (CNN) can learn sequences of convolution operations that represent image transforms in all three image domains (value, space, and frequency) and multiple levels of abstraction. Each of these convolution operations encodes a customized image filter that was learned from the training data. Furthermore, the nature of these convolution operations allowed for much deeper neural network structures than was possible with traditional artificial neural networks (ANNs). Thus, due to their deep sequential structure, CNNs can also learn hierarchical transformation and representation sequences (LeCun, Bengio & Hinton, 2015). In one single step, a DL application can start from pixels and end with a representation of the meaning of a given image (Ayyadevara & Reddy, 2020), DL also simplified the development process of CV applications: the main steps of a CV solution can now be trained. Nonetheless, DL solutions have the disadvantage of being black-box processes: the convolution operations performed by the network are learned and then coded as data in the structure of a given network, and not as separate processes, which can be isolated and individually analyzed.

Therefore, based on the urgent need to rapidly access Brazilian Funga and the image-based classification possibilities enabled by CNNs, our study aims to: (i) use our highly standardized and specialist-curated database of macrofungi occurring in Brazil (Drechsler-Santos et al., 2023) to train and validate convolutional neural networks to automatically recognize macrofungal species, and (ii) engineer a mobile application with an advanced front-end interface designed to capture images and suggest names for macrofungal species based on their images associated to a trained CNN, enabling the general public access to the Brazilian Funga while leveraging awareness and allowing citizens to contribute in knowledge construction regarding Brazilian fungal species.

Materials and Methods

This section describes our dataset and our approach to acquiring the images, building the dataset, and curating it. We will also describe the image classification CNN we have selected to develop our macrofungi image classifier, as well as the approach followed to develop the web application to host the image classifier and the image dataset. Portions of this text were previously published as part of a thesis (Chaves, 2023).

Construction of the image dataset of macrofungi

The raw image dataset (Drechsler-Santos et al., 2023) is composed of images taken by the team from the MIND.Funga/MICOLAB/UFSC research group, following the protocol by Bittencourt et al. (2022) received via email or solicited for inclusion in the dataset, where sources included partner mycologists, volunteers, citizen scientists, online mycology groups, and past research projects (Ph.D. dissertations and M.Sc Theses) focused on macrofungal species diversity. All images were submitted to a quality control and treatment pipeline, which involved three critical steps: (i) removal of poor-quality, unfocused images, and images with foreign objects (e.g., scales, human parts, knives, etc.); (ii) specimen reframing to the image center, and (iii) 1:1 aspect ratio standardization. Initially, images had their original background replaced with green and white backgrounds (HEX codes) to improve CNN training and specimen recognition. As background replacement did not significantly improve the CNN performance, new additions to the background were not modified. When multiple images of the same specimen were available, a subset that shows all specimen angles was selected. Whenever possible, several images of the same macrofungal species from different sources were incorporated into the dataset to enhance CNN predictions, accounting for contrasting scenarios (i.e.: different lighting, angles, focal length, and other image-related variables).

The dataset is organized in individual taxon directory names with current accepted scientific names for species or genera. Each taxon directory contains associated metadata composed of taxonomy, morphology, phylogeny, ecology, and distribution information.

Image classification CNN

The recognition of macrofungal specimens in the wild is a typical image classification (IC) task. In order to implement our approach, we have selected a state-of-the-art image classification CNN, the residual convolutional neural network (ResNet) (He et al., 2016). For our study, we selected a set of standard and well-proven ResNet architectures because they are available as pre-trained networks, trained on the Imagenet dataset (Deng et al., 2009) and; thus, present internal connections and weight structures that reflect image features found on most natural and common objects. CNNs pre-trained on the Imagenet dataset are much easier to train on a new data set and the procedure to use them as the starting point in a new project has become the current practice in modern computer vision projects.

In order to train the pre-trained ResNets on our macrofungi dataset, we followed the standard two-stage procedure for adapting pre-trained CNNs to a new image domain (Chaves, 2023):

Transfer learning (Stage 1): A new pair of fully-connected classification output layers, customized for the number of classes of our image domain, was generated for each network. The networks were then trained to allow learning only in the connections from the network body to its input layer and in the last two fully-connected classification layers. This first step is fast and maps the pre-trained network to the new image data.

Fine tuning (Stage 2): All connections of the networks were unfrozen and a full, much longer, training cycle was performed. This second step adapts the whole network internally to better reflect the characteristics of new image data. We additionally configured our ResNet models with an ADAM optimizer (Kingma & Ba, 2014) and the cross-entropy loss function.

In order to accelerate training and further automate the optimization of the learning rates and network momentum during training, we applied the One Cycle Policy in both training stages: transfer-learning and fine-tuning (Smith, 2018). Besides using the ResNet model, a traditional model generally seen as a first line solution that was developed employing the PyTorch-based fastai high-level DL API, we also used the state-of-the-art neural network models EfficientNet, HRnet, and Mobilenet, which were developed directly in the PyTorch framework (Chaves, 2023).

The evaluation of the models after the training was measured by the following metrics: (i) Accuracy, which measures the ratio of the number of correct predictions to the total number of predictions and is most useful for classification problems; (ii) training loss, which is the error or cost measured on the training dataset, and, for classification purpose, the cross-entropy was used; and (ii) validation loss, which was calculated on a separate dataset that was not used in training (known as the validation set). This dataset is used to simulate unseen, real-world data.

Development of the application for mobile phones

The app was developed using the Ionic framework (IONIC, 2022) with its front-end written in HTML and SCSS and the logical parts in JavaScript (AngularJS). HTML files display the elements on the screen whereas SCSS files configure the graphical properties of the elements displayed in the HTML, and TS (Type Script) files change the state and values of screen elements and communicate with the API. We have used TS files in the development process before being compiled into JavaScript files for use in the web environment. The architecture follows the MVC (model, view, and control) development pattern.

The application uses the CNN through the endpoints of the application programming interface (API) developed by de Farias (2022). The API communicates between the application and the image processing and identification part, as well as returns data stored in the application’s database, such as species data, user data, and specimen information. The API that is used by the app is divided into five endpoints: Curatorship, Species, Specimen, Specimen Submissions, and Users.

Results

In this section, we presented relevant aspects of MIND.Funga image database (Drechsler-Santos et al., 2023), the classification accuracy results with the CNN models, the online macrofungal classification system developed to host the image dataset, and the CNNs, allowing the users to classify their photographs and submit new specimens to the database.

MIND.Funga database

The raw Mind.Funga database is composed of 17,467 images of 580 macrofungal species. Nevertheless, for training CNNs, we hadto the failur to discard 3,573 images associated to 75 species due the failure in passing our first inclusive/exclusive criterium (removal of poor-quality, unfocused images, and images with foreign objects (e.g., scales, human parts, knives, etc.)). Therefore, CNN training was performed using a database with standard and high-quality of more than 13,894 images representing 505 macrofungal species. These database encompassed both morphological (Fig. 1) and taxonomic (Fig. 2) macrofungal diversities, with a mean value of 27 images per species (Fig. 3).

Figure 1 Some macrofungal species of the MIND.Funga database representing extensive morphological diversity.

(A) Gloeosoma mirabile (Berk & M. A. Curtis) Rajchenb., Pildain & C. Riquelme 2021, (B) Aurantiopileus mayensis Ginns, D. L. Lindner & T. J. Baroni 2010, (C) Amauroderma schomburgkii (Mont. & Berk.) Torrend 1920, (D) Laxitextum bicolor (Pers.) Lentz 1956, (E) Cladonia didyma (Fée) Vain. 1887, (F) Hypocrella gartneriana Möller 1901, (G) Clavaria zollingeri Lév. 1846, (H) Entoloma karstedtiae Blanco-Dios 2020, (I) Geastrum lageniforme Vittad. 1843, (J) Xylaria telfairii (Berk.) Sacc. 1882, (K) Cookeina speciosa (Fr.) Dennis 1994, (L) Entoloma flavotinctum E. Horak & Corner 1982. Photo credits (photos A–I): All the photographs (A–I) were taken by Prof. Dr. E. Ricardo Dreschler-Santos (last author of the current article).

Figure 2 Taxonomic distribution of the images associated with MIND.Funga database.

Both Ascomycota and Basidiomycota phyla are well-represented with 164 and 416 species each, from distinct genera, familes and orders.

Figure 3 Relationship between number of images and number of macrofungal species.

Each dot in the figure represents a macrofungal species and its projection in horizontal axis represents its number of images. The Box shows the values of the median and 25–75 percentiles, and other summary statistics total number of images, species, mean, min and max are depicted at the right side.

CNN models

We used four Python libraries to train the neural networks: (i) Fastai (www.fast.ai); (ii) Torch Vision (http://pytorch.org/vision/stable/index.html); (iii) Timm (http://github.com/huggingface/pytorch-image-models/tree/main/timm); and (iv) Wandb (http://wandb.ai/home).

Fastai was the base platform for the project since it defines the basic functions for training, such as (a) setting the dataset, (b) batch size, (c) transformations, and (d) fit-one-cycle. Both Torch Vision and Timm libraries were used to obtain neural network models that had not been originally implemented on Fastai platform, such as HRNet, Efficientnet, and Mobilenet. Moreover, a fourth library, Wandb, was used to plot and monitor the evaluation metrics of different CNNs, enabling visualization and comparison across various models.

The networks were trained using the MIND.Funga Project database (Drechsler-Santos et al., 2023). This dataset, which has been developing for over 3 years, stands out from many others due to its focus on species from Brazil, which were identified by mycologists who are all experts in each one of the distinct fungal taxa. MIND.Funga Project database currently contains 13,894 images, covering 505 macrofungal species. The dataset was thus divided into Training (70%), Validation (20%), and Test (10%) subsets for convolutional neural network training.

Six different networks from four different architectures were trained: (i) Resnet50, (ii) Efficientnet B4, (iii) Efficientnet B7, (iv) Hrnet18, (v) Hrnet48, (vi) MobilenetV3.

The ResNet was utilized due to its popularity and frequent use as a base network. Specifically, we selected the ResNet50 due to the complexity of the dataset, which features 505 classes (macrofungal species) and 13,894 images. The EfficientNets and HRNets were selected based on their current relevance and their proven superior performance in similar projects conducted by our research group. On the other hand, the MobileNetV3 was selected due to the need for a more efficient and lighter network, suitable for mobile applications.

The networks were trained for nine to 15 epochs, with termination based on the analysis of the Training and Validation Loss graphs (Figs. 4 and 5, respectively), aiming to reach a plateau. EfficientnetB7 architecture was prematurely halted as the Training Loss and accuracy did not generate significant results while the Validation Loss merely increased. The best performance was achieved by the EfficientnetB4, with an accuracy of 95.23%, followed by the Hrnet18 at 94.11%, and the MobilenetV3 at 94.05% (Fig. 6). The performance of the MobilenetV3 was particularly impressive compared to conventional networks since it is a CNN designed to be lightweight and efficient for the mobile market.

Figure 4 Training loss.

The X-axis represents the number of epochs, and the Y-axis the loss value. Legend: (a) Hrnet48, (b) Hrnet18, (c) EfficientnetB7, (d) EfficientnetB4, (e) MobilenetV3, (f) Resnet50-Stage2, (g) Resnet50.

Figure 5 Validation loss.

The X-axis represents the number of epochs, and the Y-axis the loss value. Legend: (a) Hrnet48, (b) Hrnet18, (c) EfficientnetB7, (d) EfficientnetB4, (e) MobilenetV3, (f) Resnet50-Stage2, (g) Resnet50.

Figure 6 Accuracy.

The X-axis represents the number of epochs, and the Y-axis the accuracy value. Legend: (a) Hrnet48, (b) Hrnet18, (c) EfficientnetB7, (d) EfficientnetB4, (e) MobilenetV3, (f) Resnet50-Stage2, (g) Resnet50.

App for mobiles

An Android Package (.apk) file, used for installing applications on Android operating systems, was generated and used for installation on Android operating systems, used by the MIND.Funga team, comprising mycologists specialized in various fungal groups, and by non-specialist guests. Furthermore, the functional requirements, based on bi-weekly meetings with the MIND.Funga research group, were also specified (Table 1):

Table 1 Functional requirements and their corresponding explanations.

Functional requirement	Detailed explanation	
RF01-log in	The application must present the “Login” menu option menu option for identifying users in the system	
RF02-identify fungus	The application must offer the menu option “What fungus is this?” menu option, so that users can take photos and identify fungi13	
RF03-show map of records	The application must present the menu option “Map of records” so that users can see the fungi identified on a map	
RF04-view species	The application must display the “Species” menu option, so that users can see information about the species registered in the system. It can sort the species alphabetically and by category	
RF05-edit species	The user can edit an already registered species, by clicking on the name of the species, if they are a curator	
RF06-register species	Register species: Curator-type users can register a new species	
RF07-proceed to curation	The application should display the menu option
“Curator”, so that users of the Curator type can collaborate with corrections and edits to the registered information.	
RF08-manage users	Users of the Administrator type will be able to
manage users via the “Manage Users” menu option. It should be possible to list all system users and edit information about these users	
RF09-show information about MIND.Funga	The application must show
the “About MIND.Funga” menu option so that users can see the application’s
version of the application and obtain information about the MIND.Funga initiative, as well as information about the project’s social networks	
RF10-show photography tips	The application must display the
“Photography tips” menu option, so that users can see the best ways to photograph fungi	
RF11-view album	Shows the specimens sent by the user with
date, substrate, observations, and location of the specimen sent	
Note:

The first column lists the functional requirements of the application for mobiles, and, in the second column, are the detailed explanations of each one of the 11 functional requirements.

Figures 7–9 depict, respectively, the cases of use, a general overview of interactions, and the interface prototypes of the app (https://mindfunga.ufsc.br/app/).

Figure 7 App: cases of use.

This figure illustrates the cases of use related to the application for mobiles.

Figure 8 App: interaction diagram.

This figure illustrates a general overview of the interactions related to the application for mobiles.

Figure 9 App: interfaces.

This figure illustrates the interfaces related to the application for mobiles. Note: As the app is initially intended to be used by a lay person in Brazil, the interface is shown in Portuguese. TRANSLATION: (order–up to down, left to right): (i) upper left panel: <which fungus is this?>… (ii) upper right panel: … (iii) lower left panel: … (iv) lower right panel: … <1. Check the framing of your image>, <2. You can insert more than one image of a same fungus>, <3. More tips?>.

Discussion

This study represents our first investigation into the world of the automated image classification of fungal specimens based upon both, in situ images and images from herbaria/fungaria collections. We applied the standard, well-accepted process for the development of image classification solutions based upon deep learning neural networks and came upon a set of neural network models that worked very well and some others that did not.

Our study has achieved a highly satisfactory performance from many convolutional neural networks for macrofungal identification, particularly the Efficientnet B4 architecture. Furthermore, the most effective CNNs were not necessarily the most complex, as the cases of EfficientnetB7 and Hrnet48, whose quality indices displayed values significantly below average. Moreover, the MobilenetV3 training outcome was a pleasantly surprising revelation since its performance mirrored that of the traditional and larger CNNs, markedly surpassing its predecessor, MobilenetV2, which had been trained during previous experiments. Therefore, our findings pointed out that the Efficientnet B4 and, to a lesser extent, MobilenetV3 CNNs were the most promising for classifying macrofungi.

EfficientNet is a family of deep neural network architectures proposed in the article (Tan & Le, 2019), designed to maximize both computational efficiency and model accuracy. This is achieved through a scaling method composed of width, depth, resolution, and efficient building blocks called MBConv. Therefore, the result is a highly efficient architecture that can be easily adapted to different sizes and tasks. There are seven models of this architecture, ranging from Efficientnet B0 to Efficientnet B7, which basically differ in network depth. Furthermore, EfficientNets use a scaling coefficient to increase the depth, width, and model resolution in a balanced way. First, a base model (EfficientNet-B0) is trained using automatic neural architecture search to find the best network architecture in terms of efficiency and accuracy. Subsequently, the base model is scaled to different sizes (EfficientNet-B1 to B7) using compound model scaling.

On the other hand, MobileNetV3, introduced by Howard et al. (2019) in a 2019 article, is the third version of the MobileNet series. The MobileNet architecture is known for its efficiency in terms of speed and size, making it ideal for use on mobile devices and other applications with limited resources. The architecture uses a combination of expansive and point convolutions to maintain efficiency, but also uses h-swish activation and h-sigmoid activation, which are optimized for mobile devices. First, the input goes through a 3 × 3 convolution layer followed by batch normalization and h-swish activation. After the initial layers, MobileNetV3 uses a series of point and expansion convolution modules and Squeeze-and-Excitation (SE) modules, which are essential for the network to focus on important features by dynamically adjusting the channel weights. The final layers of the network include a point convolution and batch normalization, h-swish activation, and a global average pooling layer. The output of this layer is then fed into a fully connected layer to produce the final output of the network.

In this study, we have tackled a problem for which there have already been ad hoc solutions freely distributed as smartphone applications to classify macrofungi. Unfortunately, the documentation about those solutions’ detailed scientific and technical background is usually unavailable. Even if it seems extremely likely that those macrofungal identification apps employ CNNs, the models they use and the precision rates they achieve, remain largely unknown. Therefore, these issues preclude a direct comparison of our results with those of many mushroom mobile apps.

Nevertheless, several ML methods have been used to classify and identify fungi over the past 5 years in literature. Although most of the mobile applications are devoted to the automatic identification of macrofungi, most of the articles in the literature are more focused on the identification of microfungi, especially those that are parasites and pathogens of both animals (including humans) and plants.

Convolutional neural networks (LeNet5, Alexnet, VGG16, SqueezeNet, ResNet, and proprietary architectures) have been explored for identifying (i) pathogenic microfungi of medical or veterinary importance (Zieliński et al., 2020; Hao et al., 2019; Prommakhot & Srinonchat, 2020; Gao et al., 2020; Liu et al., 2020; Sopo, Hajati & Gheisari, 2021; Tang et al., 2023), (ii) pathogenic or symbiotic microfungi in plants (Gaikwad et al., 2021; Marcos, Silva Rodovalho & Backes, 2019; Evangelisti et al., 2021), with quality indexes usually a few higher than 90% accuracies. Additionally, other ML methods (Decision trees/Random forest, logistic regression, k-nearest neighbor, SVM, and Bayesian networks) have been successfully used for identifying contaminant fungi on mushroom-producing farms (Rahman et al., 2022).

CNN models have also been utilized for the discrimination of distinct fungal species in taxa that comprise species complexes, both for microfungi (Billones et al., 2020) and macrofungi (Bartlett et al., 2022; Serhat, Ökten & Yüzgeç, 2023), as well as identify fungal species directly from their spores (Tahir et al., 2018). Moreover, instead of the traditional RGB image type, other ones (e.g.: spectral images) were used as the input of CNN models for identifying macrofungi (Dong et al., 2021). Besides the use of distinct image types as the input of CNN models, quite new techniques, such as Vision Transformers, have recently been applied to the task of macrofungal classification (Picek et al., 2022).

As in many other artificial intelligence (AI) systems, our CNN suffers from the closed-world assumption (CWA) problem: in a formal system of logic used for knowledge representation. CWA is the presumption that a statement that is computed as true is also known to be true and that there cannot be data outside the universe known to the system (Reiter, 1978). In an image classification CNN, this issue appears as the fact that the CNN will always classify an image, answering with its most similar class, even if what the image depicts lies completely outside the scope of the dataset used to train it. In some cases, the probability associated with the highest-ranked answer can be high, contrary to the common assumption that an image depicting a non-trained object will generate a diffuse and non-specific answer from the CNN. This suggests that simply analyzing the contrast between the CNN outputs for each class, where a significant disparity between the highest value and the others signals certainty, and a more uniform distribution indicates an unidentified object (as seen in methods replacing the Softmax function with the OpenMax function), does not assure that the image belongs to a recognized class (Bendale & Boult, 2015). Another limitation of our study is the imbalance among the classes, where some possess significantly fewer images than others, potentially leading to a bias towards the more represented species. Nonetheless, class imbalance is an intrinsic feature of biological databases (Komori & Eguchi, 2019).

Additionally, the app for mobiles was successfully developed. Several tests were performed with the members of the research team, as well as many revisions were made to the developed app to adapt it to the feedback provided by the research group members (specialists) and non-specialists, considerably improving its usability. Moreover, the suitability of the interface elements requested by the group, as well as the suitability of the use cases, raised were fully approved by all the selected users.

Conclusions

We have achieved a highly satisfactory performance from some CNN models but, by far, the most promising one was the Efficientnet B4 for macrofungi identification. Furthermore, the performance of the trained MobilenetV3 model even mirrored those of the traditional and larger CNNs. The MobileNet architecture is known for its efficiency, in terms of speed and size, making it ideal for use on mobile devices and other applications with limited resources. Using these trained CNNs, we have developed an application for mobiles that will facilitate not only the acquisition and modification of photographic data but also enable the validation and scrutiny of information about these macrofungal species. As the database is large, completely curated by experts in Mycology in various macrofungal groups, and representative of neotropical macrofungal taxa, we envisage that it will be useful amongst both experts and non-experts in Mycology, as well as a powerful educational and conservational monitoring tool. Now that we have the general framework of our research in place, we plan to focus on each of these components and go deeper into all of them, performing also a quantitative field study in order to investigate both, user acceptance and real world performance of our solution. In the field of deep learning neural networks we plan to test vision transformer models and also to reevaluate the convolutional models, testing different data augmentation strategies and also a plethora of loss functions and training strategies, besides the FitOneCycle and ADAM training strategies.

Supplemental Information

Supplemental Information 1 Image classification CNNs.

Supplemental Information 2 Application for mobile.

All the codes and documentation used for the development of the application for mobiles.

Supplemental Information 3 Justification for the inclusion of the new co-author Prof Dr Emerson Gumboski.

We would like to thank the Graduate Programmes of Microbiology (http://www.microbiologia.icb.ufmg.br/pos/) and Bioinformatics (http://www.pgbioinfo.icb.ufmg.br/) of the Universidade Federal de Minas Gerais (UFMG). We also thank the management team and directors of Conservation Units, like Parque Nacional de Aparados da Serra and Parque Nacional de São Joaquim, for permissions and support during field surveys, the Herbaria/Fungaria, like FLOR/UFSC and others for the support during access specimens, and colleagues from MICOLAB-UFSC and from other institutions for specimen collections, pictures and some helps on species identification. We would like to express our gratitude to all the collaborators of this project, which provided photos for the dataset and qualified them with as near-correct as possible names. Special thanks to Odinai Sousa for their technical support. We also extend our appreciation to Carlos J. Birckolz, Fernando Vidal, Leonardo Fusinato, Miguel Angelo Biz, Paulo R. Pezzuto, Sidney J. Damiani, through the Citizen Science Program, for their valuable assistance with photos. Additionally, we acknowledge the contributions of Aline Corrêa Ramos Marques, Ana P. Soares, Bruno Leite Tacarambi, Cassia Batista Marcon, Celia R. Dell Aquilla Balluf, Dermeval Pessin, Eder Marques, Francisco Moreira, Giuseppe Estela Dourada, Gustavo Peter Milech, Hari Borboleta Azul, Henrique H. Silveira Chapola, Janaina Santos, Jenifer Macalossi, Juliana Rogalski, Junior Gonçalves, Keinner Ouriques, Lucas Gregório, Marcos Castros, May Engel, Michelle Geraldine Campi Gaona, Renate Molz, Rodrigo Penati, Samuel Beninca, Stephan Zanga, Telmo Alexandre, Vinicius Pereira, through the social media, for providing important photos for the dataset. We are also grateful to the fungal specialists Altielys C. Magnago, Andrea Caldas, Antonio C. Sobieranski, Ariadne N. Marinho Furtado, Barbara Lima Silva, Camila Fabricio, Carlos A. Salvador Montoya, Carol Ribeiro, Cauê A. Tomaz Oliveira, Danny Newman, Denyse K. Sousa Guimarães, Diogo H. Costa-Rezende, Emanuel Grassi, Estela Rosseto, Felipe Wartchow, Filipe Pagin Cláudio, Francisco Kuhar, Gabriel Girardello, Gerardo L. Robledo, Giuliana Furci, Gonzalo Romano, Iuri Baseia, Jadergudson Pereira, Jadson D. Pereira Bezerra, Jadson J. Souza de Oliveira, José L. Bezerra, József Geml, Juli Simon Cardoso, Laise de Holanda Cavalcanti, Lara Mattana Ferst, Larissa Trierveiler Pereira, Leandro Almeida Neves, Luís A. Funez, Luiz F. Scatola, Marcel Comin, Marcela Monteiro, Marcelo D’ Aquino Rosa, Maria A. Neves, Mariana Fernandes, Mario Rajchenberg, Mateus A. Reck, Matheus Haddad, Mauro Westphalen, Melissa Palacio, Nataly Gómez Montoya, Nelson Menolli Junior, Orlando F. Popoff, Raquel Cherem Schwarz Friedrich, Renato L. Mendes Alvarenga, Roberto C. Urcelay, Rosa M. Borges da Silveira, Samuel Galvão Elias, Tattiana Gonçalves Teixeira, Valeria Ferreira Lopes, Viviana Motato-Vasquez, Weslley Ibarros Ribeiro Nardes for their invaluable assistance with taxa names.

Additional Information and Declarations

Competing Interests

Author Contributions

Data Availability

Aristóteles Góes-Neto is an Academic Editor for PeerJ.

Thiago Chaves performed the experiments, analyzed the data, authored or reviewed drafts of the article, and approved the final draft.

Joicymara Santos Xavier performed the experiments, analyzed the data, prepared figures and/or tables, authored or reviewed drafts of the article, and approved the final draft.

Alfeu Gonçalves dos Santos performed the experiments, analyzed the data, prepared figures and/or tables, authored or reviewed drafts of the article, and approved the final draft.

Kelmer Martins-Cunha performed the experiments, analyzed the data, prepared figures and/or tables, authored or reviewed drafts of the article, and approved the final draft.

Fernanda Karstedt performed the experiments, analyzed the data, prepared figures and/or tables, and approved the final draft.

Thiago Kossmann performed the experiments, analyzed the data, prepared figures and/or tables, authored or reviewed drafts of the article, and approved the final draft.

Susanne Sourell performed the experiments, analyzed the data, prepared figures and/or tables, and approved the final draft.

Eloisa Leopoldo performed the experiments, analyzed the data, prepared figures and/or tables, and approved the final draft.

Miriam Nathalie Fortuna Ferreira performed the experiments, analyzed the data, prepared figures and/or tables, and approved the final draft.

Roger Farias performed the experiments, analyzed the data, prepared figures and/or tables, and approved the final draft.

Mahatmã Titton performed the experiments, analyzed the data, prepared figures and/or tables, and approved the final draft.

Genivaldo Alves-Silva performed the experiments, analyzed the data, prepared figures and/or tables, authored or reviewed drafts of the article, and approved the final draft.

Felipe Bittencourt performed the experiments, analyzed the data, prepared figures and/or tables, and approved the final draft.

Dener Bortolini performed the experiments, analyzed the data, prepared figures and/or tables, authored or reviewed drafts of the article, and approved the final draft.

Emerson L. Gumboski performed the experiments, analyzed the data, prepared figures and/or tables, authored or reviewed drafts of the article, and approved the final draft.

Aldo von Wangenheim conceived and designed the experiments, analyzed the data, authored or reviewed drafts of the article, and approved the final draft.

Aristóteles Góes-Neto conceived and designed the experiments, analyzed the data, authored or reviewed drafts of the article, and approved the final draft.

Elisandro Ricardo Drechsler-Santos conceived and designed the experiments, analyzed the data, authored or reviewed drafts of the article, and approved the final draft.

The following information was supplied regarding data availability:

All the codes related to Convolutional Neural Networks and the development of the app for mobiles are available in the Supplemental Files.

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
