# Peer review of "Innovative infrastructure to access Brazilian fungal diversity using deep learning"

_PeerJ, doi:10.7717/peerj.17686_

## Round 0.1 · original submission · Major Revisions

· Academic Editor

Major Revisions

Dear Dr. Chaves and colleagues:

Thanks for submitting your manuscript to PeerJ. I have now received two independent reviews of your work, and as you will see, the reviewers raised some concerns about the research. Despite this, these reviewers are optimistic about your work and the potential impact it will have on research studying computational approaches for characterizing Brazilian fungi. Thus, I encourage you to revise your manuscript, accordingly, considering all of the concerns raised by both reviewers.

Importantly, please address all concerns by the reviewers regarding clarity and robustness of your analyses.

Please ensure that an English expert has edited your revised manuscript for content and clarity. Please also ensure that your figures and tables contain all the information that is necessary to support your findings and observations. The figures should also be clear and legible.

Please attempt to better discuss your research and viewpoints within the broader literature on the subjects at hand, and please ensure that critical references raised by the reviewers are included and discussed properly. Please also provide more clarity and transparency with information pertaining to your study design and comparative analyses, especially sampling protocols.

There are many comments by both reviewers that ask for more information on specific issues; please address these.

I look forward to seeing your revision, and thanks again for submitting your work to PeerJ.

Good luck with your revision,

-joe

**Language Note:** The Academic Editor has identified that the English language must be improved. PeerJ can provide language editing services - please contact us at [email protected] for pricing (be sure to provide your manuscript number and title). Alternatively, you should make your own arrangements to improve the language quality and provide details in your response letter. – PeerJ Staff

Reviewer 1 ·

Basic reporting

- For all your figures, could you enlarge the color labels? Currently, it’s a bit hard to tell which color represents which network. Could you also add a color label title to indicate that these different colors represent different networks. Or you could indicate that in your figure title.

Experimental design

- The study sample includes 17,000 photographs of 500 distinct species. Could you provide a graph to show the distribution of the number of photographs from each species? Could you provide the mean, median, and quantiles of the number of photographs from each of the 500 species? Could you provide these data for training, testing, and validation datasets separately? These could help readers understand the degree of class imbalance in your study.
- Could you discuss the generalizability of your model? How would it perform for species that are beyond the 500 species analyzed in your study?

Validity of the findings

no comments

·

Basic reporting

This manuscript mainly focuses on the development of a mobile application for Brazilian fungal diversity using convolutional neural networks (CNNs). The experiments proposed six different networks, ResNet50, EfficientNet-B4, EfficientNet-B7, HRNet18, and MobileNetv3, to train on 17,000 fungal images (covering 500 macrofungal species). The author split the dataset into three sets: training (70%), validation (20%), and test (10%). The results showed that the EfficientNet-B4 achieved the highest accuracy of 95.23%. Then, the author developed a mobile application using the best Efficient-Net-B4 model to classify the Brazilian fungal images.

Experimental design

The author only presented that the author split 17,000 images into training, validation, and test sets with a 70, 20, and 10% ratio. The author trained the CNN models with only approximately 15 epochs and achieved an accuracy of 95.23%. The author showed only the training and validation loss graphs and accuracy. However, there are many graphs that the author should visualize, such as the confusion matrix, ROC curve, etc. The author should consider the evaluation metrics, such as precision, recall, and F1-score. From my point of view, the author should do more experiments, such as data augmentation, testing the accuracy of the best model when classifying fungal images using the mobile application, computational time, etc.

Validity of the findings

The author reported only the accuracy of the best model, which is obtained from the EfficientNet-B4. The best accuracy obtained from the EfficientNet-B4 is 95.23% when training only approximately 15 epochs. Could the author show the confusion matrix and maybe show some misclassified images? Also, discussion of why the proposed method misclassified that particular images.

Additional comments

The author should be more concerned about these comments.
1. The text inside Figures 4 and 5 should be changed to English.
2. As shown in Figures 1 - 3, why does the author train the networks with only 14 epochs? The author should change the caption of Figures 1 0 3, as well.
3. In this manuscript, the author showed only one Table (see Table 1): the functional requirements. However, the author did not show any other experimental results.
4. The future work should be presented in the conclusion section.
5. Up-to-date references are required (2022-2023).
6. The author should present the dataset section and show some images of fungal. Also, please present the challenge of the Brazilian fungal dataset.
7. The theory of the CNNs is required.

---

## Round 0.2 · accepted · Accept

· Academic Editor

Accept

Dear Dr. Chaves and colleagues:

Thanks for revising your manuscript based on the concerns raised by the reviewers. I now believe that your manuscript is suitable for publication. Congratulations! I look forward to seeing this work in print, and I anticipate it being an important resource for groups studying computational approaches for characterizing Brazilian fungi. Thanks again for choosing PeerJ to publish such important work.

Best,

-joe

Reviewer 1 ·

Basic reporting

The authors have addressed my comments sufficiently.

Experimental design

The authors have addressed my comments sufficiently.

Validity of the findings

The authors have addressed my comments sufficiently.